# Genome Sequence and Phenotypic Analysis of a Protein Lysis-Negative, Attenuated Anthrax Vaccine Strain

**DOI:** 10.3390/biology12050645

**Published:** 2023-04-24

**Authors:** Lu Yuan, Dongshu Wang, Jie Chen, Yufei Lyu, Erling Feng, Yan Zhang, Xiankai Liu, Hengliang Wang

**Affiliations:** 1College of Food Science and Technology, Shanghai Ocean University, 999 Hucheng Huan Road, Nanhui New City, Shanghai 201306, China; 2State Key Laboratory of Pathogens and Biosecurity, Beijing Institute of Biotechnology, 20 Dongdajie Street, Fengtai District, Beijing 100071, China; wangdongshu@foxmail.com (D.W.); flygogo.cool@163.com (Y.L.);; 3Laboratory of Advanced Biotechnology, 20 Dongdajie Street, Fengtai District, Beijing 100071, China

**Keywords:** *Bacillus anthracis*, vaccine strain PNO2, virulence attenuation, *nprR*

## Abstract

**Simple Summary:**

The whole-genome sequencing of a putative No. II vaccine strain PNO2 with specific phenotypes was completed. The PNO2 strain is more like a Tsiankovskii strain than a Pasteur II strain. The inactivation of *nprR* gene resulted in the nonproteolytic phenotype of PNO2 and attenuated sporulation. The *nprR* was then found to be required for *Bacillus anthracis* sporulation. Further, inactivation or low expression of *abs* genes may be an important cause of vaccine strain virulence attenuation.

**Abstract:**

*Bacillus anthracis* is a Gram-positive bacterium that causes the zoonotic disease anthrax. Here, we studied the characteristic phenotype and virulence attenuation of the putative No. II vaccine strain, PNO2, which was reportedly introduced from the Pasteur Institute in 1934. Characterization of the strain showed that, compared with the control strain, A16Q1, the attenuated PNO2 (PNO2D1) was phospholipase-positive, with impaired protein hydrolysis and significantly reduced sporulation. Additionally, PNO2D1 significantly extended the survival times of anthrax-challenged mice. An evolutionary tree analysis revealed that PNO2D1 was not a Pasteur strain but was more closely related to a Tsiankovskii strain. A database comparison revealed a seven-base insertion mutation in the *nprR* gene. Although it did not block *nprR* transcription, the insertion mutation resulted in the premature termination of protein translation. *nprR* deletion of A16Q1 resulted in a nonproteolytic phenotype that could not sporulate. The database comparison revealed that the *abs* gene is also prone to mutation, and the *abs* promoter activity was much lower in PNO2D1 than in A16Q1. Low *abs* expression may be an important reason for the decreased virulence of PNO2D1.

## 1. Introduction

*Bacillus anthracis* is a Gram-positive bacterium that causes the zoonotic disease anthrax [1]. The virulence of *B. anthracis* is mainly derived from two large plasmids, pXO1 [2] and pXO2 [3], which encode the anthrax toxin and capsule, respectively. *B. anthracis* forms spores in an unsuitable environment, e.g., one with limited nutrients or moisture. Spores, which can lay dormant for decades, are the main cause of *B. anthracis* infection. Spores can germinate within macrophages and then grow into vegetative cells that form capsules and secrete toxins [4]. Both the capsules and toxins are necessary for complete virulence of *B. anthracis*. 

Vaccine development for anthrax can be traced back to the 1880s, when Louis Pasteur developed two attenuated strains [5] in France. For these strains, *B. anthracis* cultures were incubated at 42–43 °C for 12 days (Pasteur II) and 24 days (Pasteur I). Pasteur strains were used in livestock as animal vaccines until the 1930s. Pasteur’s vaccines were thought to be attenuated owing to the loss of toxin expression depending on the pXO1 plasmids [6]. However, strains with only capsules were considered to provide ineffective immunoprotection [5,7]. Scholars now widely believe that Pasteur’s vaccines were mixed cultures containing only a small percentage of fully virulent bacteria that produced toxins and improved protection [5,8,9]. Liang et al., detected three Pasteur-type strains from France, Russia, and Japan and found that all strains contained the pXO1 plasmid [10]. These authors believe that the Pasteur-type strains are attenuated because the plasmid copies are suppressed after high-temperature treatment. Studies investigating the role of pXO2 in *B. anthracis* virulence used Pasteur-type strains carrying only pXO2. Welkos et al. reported that encapsulated but nontoxigenic *B. anthracis* Pasteur 6602 strain spores were avirulent in all animals tested. Studies suggest that the increased virulence in some encapsulated but nontoxigenic strains is associated with pXO2 rather than with the amount of capsule produced [11,12,13]. Chromosomal factors may also play key roles [14].

Safety of the Pasteur strains as vaccines was unacceptable regardless of whether a small amount of a fully virulent strain was mixed in or pXO1 was incompletely cured. Therefore, after 1939, the safer and more immunogenic Sterne (pXO1^+^, pXO2^−^) vaccine strain gradually replaced the Pasteur vaccine strain. In 1940, the Soviet Union introduced the Sterne-like vaccine, ST1 [15]. In 1958, China introduced the A16R strains using ultraviolet radiation treatment [16]. Both ST1 and A16R vaccines have been used in millions of people. Protective antigen (PA), a key component of the anthrax toxin, is a major antigen that induces protective immunity against *B. anthracis* infection and has been developed into safer vaccines. 

In addition to the Pasteur-type strains lacking pXO1 and the Sterne-type strains lacking pXO2, some Pasteur strains [10,17], the Italian Carbosap strain [18], and the Russian Tsiankovskii strain [19] contain both pXO1 and pXO2 but are less virulent. The attenuation mechanism of these strains remains unknown. In-depth study of the attenuation mechanisms of these strains may enable better understanding *B. anthracis* pathogenicity and may aid in developing new vaccines.

Here, we evaluated the characteristic phenotype and virulence attenuation of the putative No. II vaccine strain, PNO2, which was reported to have been introduced into the Institute of Lanzhou Biological Products by Professor Shoushen Yang, a visiting scholar at the Pasteur Institute in 1934 [10]. PNO2 was suspected of being a Pasteur II strain. The residual pXO1 plasmid was completely cured, and the attenuated strain was named PNO2D1. This study may provide new information for better understanding the attenuation of *B. anthracis* vaccine strains.

## 2. Materials and Methods

### 2.1. Strains and Plasmids

Table 1 lists the strains and plasmids used in this study. Nontoxigenic encapsulated strain A16Q1 (pXO1^−^, pXO2^+^) was obtained from toxigenic encapsulated strain A16 using plasmid incompatibility [20]. A16Q1ΔnprR was an *nprR*-deletion strain constructed as per Wang et al. [21,22]. 

The CRISPR-Cas9 toolkit was used for pXO1 rapid curing of PNO2 [23,24]. Briefly, an N20 sequence was designed against the possible replication initiation region on pXO1 plasmids. The double-stranded N20 oligonucleotide product was inserted into pJOE8999 [25]. The demethylated scissors plasmid was then transformed into *B. anthracis* PNO2 via electroporation. The recombinant strains were grown in Luria-Bertani (LB) broth (containing 25 μg/mL kanamycin) at 30 °C (220 r/min) for 3 h. Next, 0.4% D-mannose was added to the culture for 10 h to induce Cas9 protein expression. After inducing the bacterial subculture, the plasmid-cured colonies were screened via PCR.

**Table 1 biology-12-00645-t001:** Strains and plasmids used in this study.

Plasmid or Strain	Genotype or Description	Source
Plasmid		
pBE2	Shuttle vector, Kanr in *B. anthracis* and Ampr in *E. coli*	[21]
pBE2nprR	pBE2A carrying nprR gene, nprR complementation plasmid, Ampr in *E. coli*, Kanr in *B. anthracis*	This study
pHT304	Shuttle vectors, Ermr, Ampr	[26]
pHT304-lacZ	Promoterless lacZ vector, Ermr, Ampr, 9.7 kb	[27,28]
pHT304-Pabs	pHT304-lacZ carrying Promoter Pabs, Ampr in *E. coli*, Ermr in *B. anthracis*	This study
pHT304-Pabsm	pHT304-lacZ carrying mutant promoter Pabsm, Ampr in *E. coli*, Ermr in *B. anthracis*	This study
*E. coli*		
DH5α	F2, Q80d/lacZDM15, D(lacZYA-argF)U169, deoR, recA1, endA1, hsdR17(rk 2,mk+), phoA, supE44l2, thi-1, gyrA96, relA1	Transgen, Beijing, China
JM110	rpsL(StrR), thr, leu, endA, thi-1, lacy, galK, galT, ara, tonA, tsx, dam−, dcm−, supE44(lac-proAB), F- [traD36, proAB, lacIqlacZΔM15]	Transgen, Beijing, China
*B. anthracis* strain		
A16Q1	Nontoxigenic encapsulated strain; pXO1−, pXO2+	[20]
A16Q1ΔnprR	A16Q1 nprR deletion mutant	This study
PNO2	Putative No.II Vaccine strain; pXO1+, pXO2+	[10]
PNO2D1	PNO2 strain cured pXO1; pXO1−, pXO2+	This study
PNO2D1nprRc	nprR complementation strain containing pBE2nprR plasmid; Kanr	This study
Pabs/PNO2D1	PNO2D1 strain containing plasmid pHT304-Pabs, Ermr in *B. anthracis*	This study
Pabs/A16Q1	A16Q1 strain containing plasmid pHT304-Pabs, Ermr in *B. anthracis*	This study
Pabsm/A16Q1	A16Q1 strain containing plasmid pHT304-Pabsm, Ermr in *B. anthracis*	This study

### 2.2. Phenotypic Characterization

All *B. anthracis* strains were grown at 37 °C with shaking at 220 r/min in NaCl-free LB medium (containing only 10 g/L tryptone and 5 g/L yeast extract) [29]. After 12 h of culturing, spore staining (malachite green and safranin O) [28] and Gram staining (Gram Stain Kit, Hopebio, Shandong, China) were performed to observe the microscopic morphology of the vegetative cells. After 24 h, the spores were restained to observe their formation. The NaCl-free LB medium was additionally supplemented with 0.9% NaHCO_3_ (*m*/*v*) and 5% horse serum (*v*/*v*) to promote capsule formation. After culturing for 12 h in 5% (*v*/*v*) CO_2_, Indian ink staining was performed to visualize capsules. 

Hemolysis, phospholipase, and proteolysis of the strains were observed using specific plates. The extracellular protease activity phenotypes were investigated on milk agar plates [21,30]. Hemolysis of the *B. anthracis* strains was tested on Colombia blood agar [31]. The phospholipase phenotypes were tested on egg yolk agar plates (LB agar supplemented with 1% [*v*/*v*] egg yolk) [24]. Fresh cultures (10 µL) were pipetted onto these plates and cultured at 37 °C for approximately 12 h.

Sporulation efficiency was determined as previously described [28]. Strains were cultured in NaCl-free LB medium at 37 °C for 48 h and then heat-inactivated (70 °C for 30 min). The number of spores was determined using heat-resistant colony-forming units (CFUs) on LB agar plates. The sporulation rate was determined by the ratio of the number of spores to the total number of viable cells.

### 2.3. Genome Sequencing

High-quality genomic DNA was extracted using a Wizard Genomic DNA Purification Kit (Promega, Madison, WI, USA). Whole-genome sequencing was performed using the Illumina NovaSeq 6000 platform and PacBio Sequel II system (Frasergen Bioinformatics Co., Ltd., Wuhan, China). The PacBio reads were de novo assembled using Microbial Assembly (smrtlink8), HGAP4 [32] and Canu (v.1.6) [33] software. The assembled genome sequence was further polished with Illumina data using Pilon v1.22 (https://github.com/broadinstitute/pilon/ accessed on 21 March 2023). Annotation was performed using the NCBI Prokaryotic Genome Annotation Pipeline (PGAP) [34]. 

The Parsnp tool from Harvest Suite was used for rapid calling of the single nucleotide polymorphism (SNP) positions [35]. For this, a chromosome dataset, representing genomes from public databases, and the newly sequenced *B. anthracis* strain, PNO2D1, were aligned against the *B. anthracis* chromosome “Ames Ancestor” (NC_007530) as a reference strain using Parsnp (parsnp -r -d -c). To export the identified SNP positions, Harvest Tools (version 1.0) from the same software suite was used to create a variant calling file (VCF) listing all SNP positions. The SNPs in the purified VCF of each strain were concatenated into a single sequence; then, all single sequences were combined to form a new multiple-sequence fasta format file. This concatenated multiple-sequence file was used as input for MEGA11 to compile a multiple-sequence alignment and to construct a maximum-likelihood tree [36], which was further edited using itol tools (https://itol.embl.de./ accessed on 27 March 2023) online. This tree was rooted at the midpoint.

### 2.4. Virulence Testing

The purified spores were counted via the gradient dilution method. The prepared spores were diluted over a gradient using ultra-pure water. Female DBA/2 mice (6–8 weeks old) were purchased from Beijing Vital River Laboratory (Beijing, China). The A16Q1 and PNO2D1 spores were diluted to 1 × 10^5^ CFU/mL. The mice (n = 10 per group) were inoculated intraperitoneally with a 100 μL volume of spores. The survival time of the mice was recorded until all mice were dead. The log-rank test was used for survival analysis [37].

### 2.5. Quantitative Reverse-Transcription PCR 

A qRT-PCR analysis was performed to identify the differences in gene expression between wild-type and mutant strains. Total RNA was extracted from *B. anthracis* cells cultured using a Bacterial RNA Extraction Kit (Vazyme, Nanjing, China). Double-stranded cDNA was generated using a HiScript II Q RT SuperMix for qPCR (+gDNA wiper) (Vazyme, Nanjing, China). A qPCR analysis was performed using Taq Pro Universal SYBR qPCR Master Mix (Vazyme) and the CFX96 Connect Real-Time PCR System (BioRad; Hercules, CA, USA). Table 2 lists the primers used for the qRT-PCR. The relative changes in gene expression were measured using the ΔΔCt method with *gatB_Yqey* [38] as the reference gene.

### 2.6. β-Galactosidase Assays

*B. anthracis* strains containing *lac*Z transcriptional fusions were grown in liquid NaCl-free LB medium at 37 °C for 10 h. The culture samples (1.5 mL) were collected into 0.5 mL of the Z buffer (0.06 M Na_2_HPO_4_, 0.04 M NaH_2_PO_4_, 0.01 M KCl, 1 mM MgSO_4_, and 1 mM DTT). The β-galactosidase activities of the samples were measured as previously described [39] and expressed as Miller units per mg of protein. Briefly, the cells were lysed using the Precellys24 (Bertin Technologies, Paris, France). Subsequently, 0.7 mL of the Z buffer and 200 μL of ONPG (2-nitrophenyl-beta D-galactoside, 4 mg/mL, Sigma, St Louis, MO, USA) were added to 100 μL of the lysate supernatant. The mixture was incubated at room temperature, and the reaction was stopped by adding 0.5 mL of 1M Na_2_CO_3_. The optical density of the reaction mixture was then measured at 420 nm. At least three independent cultures were assayed for enzyme activity.

## 3. Results

### 3.1. Phenotypic Characteristics of PNO2D1

We examined several PNO2D1 phenotypes. Sporulation, bacterial morphology, lecithinase, and proteolysis exhibited unique phenotypes. The sporulation ability of PNO2D1 appeared much weaker than that of A16Q1. At 24 h, A16Q1 was nearly fully sporulated, whereas PNO2D1 was still mostly in the vegetative state (Figure 1A). Both the A16Q1 and PNO2D1 strains had capsules. The A16Q1 vegetative cells formed typical long chains, whereas the PNO2D1 vegetative cells formed relatively independent rods (Figure 1B). A16Q1 and PNO2D1 were both hemolysis-negative on sheep blood agar, and PNO2D1 was phospholipase-positive on egg yolk agar plates. On milk agar plates, the A16Q1 colonies had clear proteolysis rings surrounding them; the PNO2D1 colonies did not. This suggested that PNO2D1 lacked extracellular protease activity (Figure 1C).

The DBA2 mice were challenged with 10^5^ CFUs of spores, and the capsular strains showed strong virulence. The PNO2D1 virulence was significantly weaker than that of A16Q1; however, PNO2D1 still killed all mice after 2 days (Figure 1D).

### 3.2. Whole-Genome Sequencing of PNO2 and PNO2D1

In total, 3,818,380 reads were generated for PNO2, and 3,996,493 reads were generated for PNO2D1. The PNO2 chromosome was 5,229,474 bp (GC content: 35.28%); the PNO2D1 chromosome was 5,229,500 bp (GC content: 35.38%). The PNO2 plasmids were 181,764 bp (pXO1) and 94,725 bp (pXO2); the PNO2D1 pXO2 plasmid was also 94,725 bp. The average sequencing depth was 890.47× for the PNO2 chromosome, 2484.66× for plasmid pXO1, and 740.16× for pXO2. The average sequencing depths were 764.75× for the PNO2D1 chromosome and 619.53× for pXO2. All whole-genome coverages reached 100%. The sequencing results showed that plasmid pXO1 was completely cured in PNO2D1. The PNO2D1 chromosome contained fewer mutations than did the PNO2 chromosome. The genomes were annotated using the NCBI PGAP. The GenBank assembly accession numbers for the genomes are GCA_022014755.1 for PNO2 and GCA_022014775.1 for PNO2D1. 

The CanSNP typing results showed that the Sterne-type vaccine strains belonged to A.Br.001/002 and the Pasteur-type vaccine strains belonged to A.Br.008/009. PNO2D1 belonged to A.Br.008/009, which is the same group that Cvac02 belongs to (China Veterinary Anthrax Spore Vaccine, strain No. 2). Interestingly, the genome-wide SNP genotyping results showed that both Cvac02 and PNO2D1 were more closely related to Tsiankovskii strains than to Pasteur strains ATCC240 and ATCC4728 and Smith strains (Figure 2A). 

PNO2 was reported to have been introduced into China in 1934 from the Pasteur Institute and presumed to be a Pasteur II strain. In August 2012, an anthrax outbreak in Liaoning Province, Northeast China, infected seven persons and killed dozens of cows. Cvac02 was the local veterinary vaccine strain (NCBI accession PRJNA252785) that was reported to be Rentian II introduced from Japan [10]. However, a genome-wide SNP clustering analysis showed that PNO2 and Cvac02 were more likely unique Tsiankovskii strains because they contained both virulence plasmids of *B. anthracis* and produced both capsules and PA (Figure 2A). Cvac02 was successfully used to immunize livestock in the Soviet Union until 1960. 

Liang et al., reported that the Pasteur II, Tsiankovskii II, and Rentian II strains all contained two plasmids [6,10]; however, our genome-wide analysis suggested that these strains might have originated from the Tsiankovskii strains. The full-sequence SNP clustering analysis results for pXO1 of the Pasteur II strain reported by Liang et al., (GenBank: gb|KT186230.1) showed that it was also similar to pXO1 of the Tsiankovskii strain (Figure 2B).

### 3.3. nprR Mutation Resulted in a Nonproteolytic Phenotype and Decreased Sporulation of PNO2D1

The PNO2 and PNO2D1 genomes each contained a 7-base insertion mutation (CGAAAAT) within the *nprR* gene position between bases 123 and 124 (locus GBBA0597 in reference genome GCF_000008445.1; Figure 3A). The quantitative PCR results showed that this insertion mutation did not negatively affect *nprR* transcription (its expression was higher; Figure 3B). However, translation of the encoded mRNA yielded an inactive prematurely terminated NprR protein. Therefore, the transcription level of the neutral protease Npr599, regulated by NprR, was very low. This resulted in negative proteolysis of PNO2D1 (Figure 3C). Furthermore, *nprR* knockout in A16Q1 [21] showed that extracellular protease activity was greatly reduced in the deletion mutant A16Q1ΔnprR (Figure 3C). 

A16Q1ΔnprR could not form spores, suggesting that NprR might be necessary for sporulation in *B. anthracis*. After culturing for 48 h, the proportion of heat-resistant spores in all viable bacteria was only 5.05% in PNO2D1 compared with 30.45% in A16Q1 (*t*-test, *p* < 0.01). The sporulation rate for PNO2D1 containing complementary *nprR* (PNO2nprRC) was significantly increased (48.33%, *t*-test, *p* < 0.01; Figure 3D). 

The *nprR-npr599* gene cluster was analyzed in most vaccine strains mentioned in the NCBI database (Table 3). *nprR*, which affects expression of the NprR/NprX quorum-sensing regulon active in bacilli, showed a high mutation ratio. The FDAARGOS_341, Sterne, and Brazilian vaccinal strains contained no *nprR* mutations (Table 3). PNO2D1 and the Tsiankovskii strain contained the seven-base insertion mutation (CGAAAAT) in *nprR*. This is consistent with the results of our genomic clustering analysis, indicating that the PNO2 and Tsiankovskii strains are highly similar. Additionally, the Pasteur strains have an insertional mutation “A” base in *nprR* and, thus, may yield a nonproteolytic phenotype. *nprR* sequencing may enable quickly distinguishing Tsiankovskii-type strains from Pasteur-type strains.

We continuously passaged PNO2 at high temperatures to clarify whether high-temperature treatment during plasmid curing resulted in *nprR* mutants. However, 50 passages yielded no protease-defective colonies. 

### 3.4. abs Gene Cluster Expression May Be Associated with PNO2D1 Attenuation 

To determine why PNO2D1 exhibits attenuated virulence, we analyzed *B. anthracis* virulence-related genes from the Virulence Factor Database (http://www.mgc.ac.cn/VFs/main.htm accessed on 22 February 2023). We found no mutations that might directly reduce virulence. pXO2 copy numbers detected via qPCR showed that the plasmid copy numbers were significantly lower for PNO2D1 than for A16Q1 (Figure 4B, *p* < 0.05, *t*-test), which was consistent with previous research [10]. 

We analyzed genes with high mutation rates similar to *nprR* and then focused on the *abs* gene cluster. Table 3 lists the *abs* gene clusters of most vaccine strains in NCBI. In Cvac02, a base insertion mutation after 56 bases (56_57insA) in the open reading frame of *absA* led to a nonsense mutation from TGT to TGA. Strains ATCC4728, ATCC240, and Smith1013 contained 51 base deletions 58 bp upstream from the *absA* start codon, ATG (Figure 4A, Table 3). The 51-base deletion promoter P_absm_ had very low enzyme activity on β-galactosidase assays (Figure 4C), suggesting that this mutation in the −10 and −35 regions inactivated the *absA* promoter and likely significantly affected expression of the *abs* gene cluster. The *abs* gene clusters of PNO2 and PNO2D1 were not mutated compared with those of the reference strain, Ames. qPCR revealed that transcriptional levels of the *absA* and *absE* gene clusters were downregulated in PNO2D1 (Figure 4D). This was also demonstrated in the β-galactosidase assays (Figure 4C). Low expression of the *abs* siderophore gene cluster may affect PNO2D1 virulence [40]. PerP was reported to be a positive regulator of the *abs* gene cluster [41]. PerP expression was downregulated at the mRNA level, which may have led to the low *abs* expression.

## 4. Discussion

Live attenuated anthrax vaccines were prepared using plasmid-cured strains. We screened two types of vaccine strains: pXO1-cured stains represented by Pasteur strains, which produced no toxins, and pXO2-cured stains represented by Sterne strains, which produced no capsules. Previous studies have shown that the anthrax capsule provides ineffective immunoprotection, suggesting that the Pasteur strain may be a poor vaccine strain. However, Pasteur strains have long been used as effective veterinary vaccines. Later studies found that some Pasteur strains contained two plasmids, pXO1 and pXO2, but they were attenuated. Some Pasteur strain vaccines were mixed with a few virulent strains, thus providing immunoprotection. However, neither of these conditions ensured that the Pasteur strain was safe; thus, it was gradually replaced by the Sterne strain.

The PNO2 strain, preserved in China, contains two virulence plasmids. Here, the plasmid pXO1 of PNO2 was cured using the CRISPR-Cas9 system tool and named PNO2D1. PNO2 was documented to be introduced from the Pasteur Institute in France and was therefore thought to be a possible Pasteur II strain. However, we analyzed the genome-wide SNPs of PNO2 and found that PNO2 was more likely derived from the Tsiankovskii I strain from the former Soviet Union. An SNP analysis of pXO1 showed that the Pasteur II strain reported by Liang et al. differed from PNO2 and was more similar to the Tsiankovskii strains than to other strains. The Pasteur II, Tsiankovskii II, Rentian II, and Cvac02 strains have high similarity and likely originated from the same strain. Owing to the age, technology, and high similarity within *B. anthracis* strains, any erroneous records of a strain could not be detected or corrected for a long time. With advances in genome-sequencing technology, we introduced a clustering analysis of whole-genome SNPs into the strain identification of *B. anthracis* sets, enabling more accurate strain analysis and traceability. This allows for clarifying historical strains, correcting errors, and further promoting vaccine development.

The PNO2 strains had some unique phenotypes; thus, we performed sequencing to explain these phenotypes. *nprR* had one notable mutation. *nprR* is a spore-related regulatory gene in the *Bacillus cereus* group, and it forms a quorum-sensing system with NprX. Related studies with *B. thuringiensis* showed that *nprR* regulates expression of the downstream protease Npr599. A study reported that incubating Sterne colonies of *B. anthracis* on LB agar at 30 °C for prolonged periods yielded many mutants negative for extracellular protease activity, with *nprR* exhibiting different mutational variants [21,30]. PNO2D1 was deficient in extracellular protease activity and had difficulty with sporulation. An insertion mutation in *nprR* in PNO2D1 caused premature termination of translation. *nprR* inactivity leads to the absence of Npr599 expression and a protease-negative phenotype. We complemented *nprR* through plasmid pBE2 so that PNO2D1 obtained a proteolytic loop and simultaneously greatly increased the rate of spore formation. *nprR* knockout in the control strain, A16Q1, resulted in deficient sporulation; thus, NprR may be necessary for *B. anthracis* sporulation. In *B. thuringiensis*, NprR is thought to inhibit spore formation, and to regulate the bacteria to enter saprophytic growth or to form spores by cooperating with spo0A. However, no published studies have reported whether *nprR* deletion affects the sporulation rates in other *B. cereus* group members. Therefore, the function of NprR in *B. cereus* requires further study. Furthermore, when we compared the *nprR* genes of vaccine strains in the NCBI database, we found that most vaccine strains had inactivated *nprR* genes; however, *nprR* had no detectable mutations in our serial 50-passage test. This high mutation frequency may be due to the selection of vaccine strains. NprR deletion was screened, which may have reduced the degradation of major extracellular antigen proteins, such as PA, and improved the immunity effects.

PNO2D1 showed attenuated virulence, and previous studies suggested that this may be due to decreased plasmid copy numbers. However, we screened strain ChBA30D, which naturally lost pXO1 in a ChBA30 culture [31]. ChBA30D’s plasmid copy numbers did not significantly differ from those of PNO2D1, but its virulence was similar to that of A16Q1. Thus, the decreased plasmid copy numbers may not be the only reason for the reduced virulence of PNO2D1. We also found low expression of the petrobactin synthesis gene cluster in PNO2D1. The petrobactin synthesis gene cluster, *abs*, is closely related to virulence in *B. anthracis*, and *absA* deletion (GBBA1981) results in avirulence [40]. Therefore, we speculate that low *absA* expression may be one reason for the weakened virulence of PNO2D1. Additionally, some Pasteur strains had *absA* mutations, which may be important reasons for their attenuation. Unfortunately, we have not found a typical Pasteur strain and could not perform further experimental validation.

## 5. Conclusions

We completed whole-genome sequencing of PNO2. PNO2 is more similar to Tsiankovskii strains than to Pasteur strains, and a nonsense mutation in *nprR* resulted in the nonproteolytic phenotype and decreased sporulation in PNO2. Herein, we provide the first known experimental evidence that *nprR* is required for *B. anthracis* sporulation. We also searched the NCBI database for mutations in *nprR* and the *abs* gene cluster in vaccine-associated strains. Inactivation or low expression of *abs* may be an important cause of vaccine strain attenuation.

## Figures and Tables

**Figure 1 biology-12-00645-f001:**
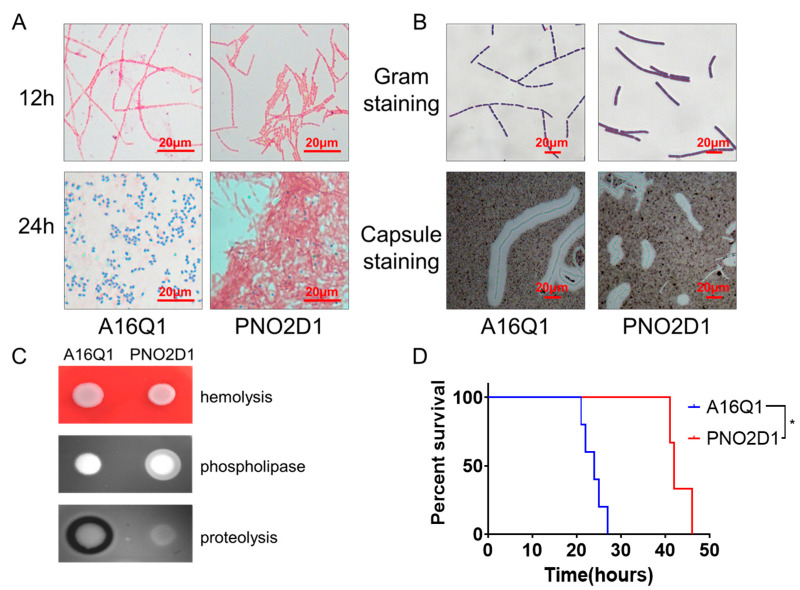
Phenotypic characteristics of PNO2D1 strains. (**A**) Schaeffer-Fulton spore staining for A16Q1 and PNO2D1 strains at 12 and 24 h. (**B**) Gram and capsule staining for A16Q1 and PNO2D1 strains. (**C**) Hemolysis, phospholipase, and proteolysis for A16Q1 and PNO2D1. (**D**) Survival curve of DBA2 mice undergoing virulence challenge (log-rank test, * *p* < 0.05).

**Figure 2 biology-12-00645-f002:**
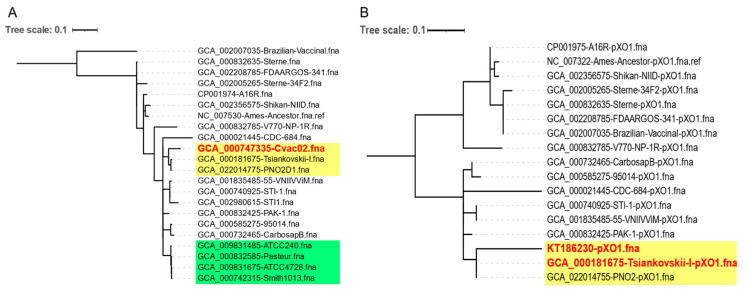
Genome SNP genotyping of PNO2 strains. (**A**) Genome-wide SNP clustering analysis using Parsnp tools. PNO2D1 was closer to Tsiankovskii strains (yellow) than to Pasteur strains (green). (**B**) SNP clustering analysis of pXO1 plasmids using Parsnp tools. Tsiankovskii strains are highlighted in yellow. KT186230.1 was the pXO1 sequence of the reported Pasteur II strain. No Pasteur strains from NCBI were included because they lacked pXO1.

**Figure 3 biology-12-00645-f003:**
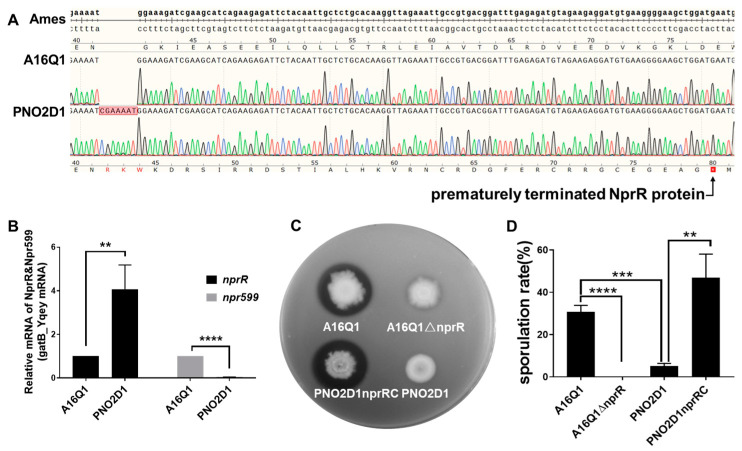
Nonproteolytic phenotype and decreased sporulation of PNO2D1. (**A**) Sequencing verification of *nprR* insertion mutations in the genome of PNO2D1. Ames represents a partial *nprR* sequence of the Ames Ancestor strain, which was used as a reference sequence. The Sanger sequencing results for A16Q1 and PNO2D1 were compared with the reference sequence. The *nprR* sequence for A16Q1 was consistent with the reference sequence. The PNO2D1 sequence contained a 7-base insertion between bases 123 and 124, resulting in early termination of translation at amino acid position 80. (**B**) Relative *nprR* and *npr599* mRNA levels in A16Q1 and PNO2D1. Relative to A16Q1, the *nprR* mRNA level was upregulated in PNO2D1 (4.08-fold, *t*-test, ** *p* < 0.01), whereas the *npr599* mRNA level was near zero (0.04-fold, *t*-test, **** *p* < 0.0001). (**C**) Proteolysis of PNO2D1 and A16Q1. Deletion mutations of *nprR* in A16Q1 and insertion mutation of *nprR* in PNO2D1 resulted in the nonproteolytic phenotype. Wild-type *nprR* complemented PNO2D1 proteolysis. (**D**) Sporulation rates of PNO2D1 and A16Q1. Deletion mutations in *nprR* in A16Q1 and an insertion mutation in *nprR* in PNO2D1 resulted in decreased sporulation compared with that of A16Q1 (*t*-test, **** *p* < 0.0001, *** *p* < 0.001). Wild-type *nprR* complemented sporulation in PNO2D1 (*t*-test, ** *p* < 0.01).

**Figure 4 biology-12-00645-f004:**
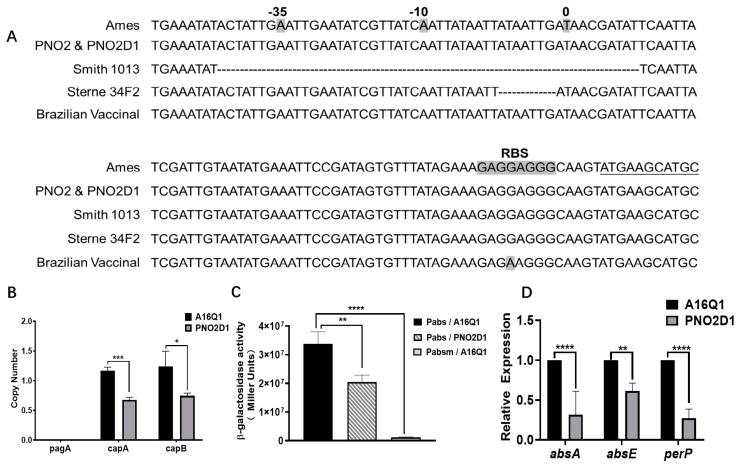
Expression of the *abs* gene cluster may be associated with PNO2D1 attenuation. (**A**) Schematic representation of mutations in the promoter region upstream from the *absA* genes. “0” represents the transcription start site. The −10 and −35 regions are located upstream from this site. The Pasteur-type strain, represented by Smith 1013, had a deletion mutation at positions −108 to −59 upstream of the transcription start site. Sterne 34F2 also had a deletion mutation at positions −75 to −69. RBS indicates a ribosome-binding site, and the AGGAGG sequence of the Brazilian vaccine strain was mutated to AGAAGG. However, the *abs* gene clusters in PNO2 and PNO2D1 were not mutated compared with those of the reference sequence. (**B**) Plasmid copy numbers of PNO2D1 and A16Q1. (*t*-test, *** *p* < 0.001, * *p* < 0.05) (**C**) β-galactosidase assays of the *abs* and mutated *abs* promoters. Wild-type *abs* promoter in PNO2D1 showed lower enzyme activity on β-galactosidase assays (*t*-test, ** *p* < 0.01). The 51-base deletion *abs* promoter, referenced to Pasteur-type strains, showed an almost complete loss of enzymatic activity on β-galactosidase assays (*t*-test, **** *p* < 0.0001). (**D**) Relative mRNA levels of *absA*, *absE*, and *perP* in A16Q1 and PNO2D1.

**Table 2 biology-12-00645-t002:** qRT-PCR primers used in this study.

Primers	Sequence (5′–3′)	Note
q1981-F	AGGAAGTGGAGAATGGATTACAG	qPCR analysis for *absA*
q1981-R	GCCGATAACCGACAGCATA	
q1985-F	CGACAATGAACTGACTATGGA	qPCR analysis for *absE*
q1985-R	TTACTGCCGCTACAAGAC	
qPerR-F	GCAACTGTCTATAATAACTTAC	qPCR analysis for *perR*
qPerR-R	ATCAACAATCTTACCACATT	
qNprR-F	TTGTTCTGTCTCATACTTA	qPCR analysis for *nprR*
qNprR-R	TATATGCGTTCTACTTGT	
qNpr599-F	AGATGCTCACTACTATGC	qPCR analysis for *npr599*
qNpr599-R	CAATTCCACCAGACAATG	
gatB_Yqey-F	AGCTGGTCGTGAAGACCTTG	reference
gatB_Yqey-R	CGGCATAACAGCAGTCATCA	

**Table 3 biology-12-00645-t003:** Mutation information for *nprR* and *abs* in vaccine-related strains from the NCBI database.

GenBank Accession	BioSample	Assembly	Geographic Location	Strain	References	Mutation of nprR *	Mutation of abs *
GCA_001835485.1	SAMN05915713	ASM183548v1	Georgia: Tbilisi	55-VNIIVViM	PMID: 28007853	1064_1065insCTTAG	NA
GCA_002980615.3	SAMN08627228	ASM298061v3	Russia: Saratov	STI1	PMID: 9413092	1064_1065insCTTAG	NA
GCA_000740925.2	SAMN02839456	ASM74092v2	Russia	STI-1(STI-89/K2789)	PMID: 25237016	1064_1065insCTTAG	NA
GCA_022014775.1	SAMN25418999	ASM2201477v1	China: Beijing	PNO2D1	PMID: 27029580	123_124insCGAAAAT	NA
GCA_000181675.2	SAMN02436236	ASM18167v2	Soviet Union	Tsiankovskii I	PMID: 33962043	123_124insCGAAAAT	NA
GCA_000832785.1	SAMN03092715	ASM83278v1	USA: Florida	V770-NP-1R	PMID: 25931591	210_214GGATG	NA
GCA_000747335.1	SAMN02898388	ASM74733v1	China: Liaoning	Cvac02	PMID: 27299730	245delC	absA 56_57insA
GCA_000512775.2	SAMN02641484	ASM51277v2	China	A16R	PMID: 26116813	NA	NA
GCA_000732465.1	SAMN02910129	ASM73246v1	Italy	Carbosap	PMID: 23405332	319_325delGATTTTG	absA 511C>T
GCA_000585275.1	SAMN02951914	NA	Italy	95014	NA	319_325delGATTTTG	absA 511C>T, absB 308_309insT
GCA_009831675.1	SAMN12620930	ASM983167v1	USA	ATCC 4728	PMID: 31896628	431_432insA	absA −108_−59del **
GCA_009831485.1	SAMN12620928	ASM983148v1	USA	ATCC 240	PMID: 31896628	431_432insA	absA −108_−59del
GCA_000742315.1	SAMN02732407	ASM74231v1	USA	Smith 1013	PMID: 25301645	431_432insA	absA −108_−59del
GCA_000832585.1	SAMN03024436	ASM83258v1	Unknown	Pasteur BBG	PMID: 25931591	431_432insA	absA −108_−59del
GCA_002356575.1	SAMD00026520	ASM235657v1	Japan:Tokyo	Shikan-NIID	PMID: 26089418	873delA	NA
GCA_002007035.1	SAMN06270273	ASM200703v1	Brazilian	Brazilian Vaccinal	PMID: 28807610	NA	absA −10G>A, absB 1278T>C
GCA_002208785.2	SAMN06173354	ASM220878v2	USA:MD	FDAARGOS_341	PMID: 31346170	NA	absB 1278T>C
GCA_000832635.1	SAMN03010431	ASM83263v1	USA	Sterne	PMID: 25931591	NA	absB 1278T>C
GCA_000832425.1	SAMN03010430	ASM83242v1	Pakistan	PAK-1	PMID: 25931591	NA	NA
GCA_002005265.1	SAMN06161234	ASM200526v1	not collected	Sterne 34F2	PMID: 21673962	NA	absA −75_−69delATAATT, absB 1278T>C
GCA_000021445.1	SAMN02603931	ASM2144v1	Unknown	CDC 684	PMID: 21962024	NA	absC 506_507insG

NA: not available. * Reference to Ames Ancestor strain. Mutation description reference to HGVS Recommendations for the Description of Sequence Variants: 2016 Update. ** Fifty-one nucleotide bases ACTATTGAATTGAATATCGTTATCAATTATAATTATAATTGATAACGATAT were deleted at −108 to −59 upstream from the *absA* start codon.

## Data Availability

The data presented in this study are openly available in NCBI Assembly repository at https://www.ncbi.nlm.nih.gov/data-hub/genome/GCF_022014775.1/, reference number ASM2201477v1.

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
