# Peer review of "Genome Sequence and Phenotypic Analysis of a Protein Lysis-Negative, Attenuated Anthrax Vaccine Strain"

_biology, 2023, doi:10.3390/biology12050645_

Round 1

Reviewer 1 Report

The topic of the article alludes to an interesting investigation that concerns the sequence and phenotypic analysis of an Anthrax vaccine strain, accompanied by two depositions in GenBank, and could be potentially published in Biology.

While the methods followed in this work are suitable for this kind of analysis and the experimental section is well performed, the text needs some modification. In particular: a) there is a very short description of methods that were followed in the present work (with the proper citations) in the text and b) the figure legends luck of crucial description. Thus, the manuscript must be enhanced with more information and experimental details.

These are some notes, which might improve the quality of the manuscript.

1. Some typo errors throughout the text must be addressed, i.e. use italics for Bacillus anthracis in line 41, erase lines 157-160 from template, change “se-quenc33ing” to “sequencing” in line 189, 

2. line 17: “The PNO2 strain is more like a Tsiankovskii strains than” to “The PNO2 strain is more like a Tsiankovskii strain than”

3. line 80: There is no reference of Table 1 in the text.

4. lines 82-83: Authors could be more explicit here.

5. line 125: More details about the constructed tree must be provided.

6. line 136: A citation for logrank test is the following: 10.1136/bmj.328.7447.1073

7. line 175, Figure 1A: The scale at the bottom of the image is indistinct.

8. line 258, Figure 3A: A more representative legend is needed.

9. line 289, Figure 4A: A more representative legend is needed for the alignment and the illustrated elements.

10. lines 302-303: “Early anthrax vaccine … of the plasmid.” Rephrase for clarity.

11. line 394: Extensive editing of references’ format is needed in order to comply with journal’s requirements.

Author Response

Responses to Reviewer 1

Thank you for your letter and comments concerning our manuscript. The comments were valuable and helpful for revising and improving our paper. We have reviewed the comments carefully and made the necessary corrections. The revised portions are in red font in the revised manuscript. The responses to the comments are as follows.

Point 1: Some typo errors throughout the text must be addressed, i.e. use italics for Bacillus anthracis in line 41, erase lines 157-160 from template, change “se-quenc33ing” to “sequencing” in line 189,

Response 1: Thank you for carefully reading our manuscript. We apologize for these errors. We corrected the errors and have checked and proofread the entire text. The manuscript has been edited by a native English-speaking, board-certified editor in the life sciences.

Point 2: line 17: “The PNO2 strain is more like a Tsiankovskii strains than” to “The PNO2 strain is more like a Tsiankovskii strain than”

Response 2: We corrected this error.

Point 3: line 80: There is no reference of Table 1 in the text.

Response 3: Thank you. Table 1 is now referenced in line 85 of the manuscript.

Point 4: lines 82-83: Authors could be more explicit here.

Response 4: Thank you for your suggestion. The method used to construct these two strains has now been updated with the CRISPR-cas9 toolkit. After publication of our work on the Bacillus anthracis CRISPR-Cas9 toolkit, the importance of these methods decreased. The CRISPR-Cas9 toolkit is succinctly described in lines 88–96. Here, we added two more detailed references (refs.21 and 22) so that interested readers can follow up.

Point 5: line 125: More details about the constructed tree must be provided.

Response 5: Thank you for your suggestion. We added the relevant content to subsection 2.3 (lines 130-141) in the revised manuscript.

Point 6: line 136: A citation for logrank test is the following: 10.1136/bmj.328.7447.1073

Response 6: Thank you. We cited this reference (line 150).

Point 7: line 175, Figure 1A: The scale at the bottom of the image is indistinct.

Response 7: Thank you. We updated Figure A to show better details.

Point 8: line 258, Figure 3A: A more representative legend is needed.

Response 8: Thank you. We revised the legend as per your suggestion (lines 267–274).

Point 9: line 289, Figure 4A: A more representative legend is needed for the alignment and the illustrated elements.

Response 9: Thank you. We revised the legend as per your suggestion (lines 305–311).

Point 10: lines 302-303: “Early anthrax vaccine … of the plasmid.” Rephrase for clarity.

Response 10: Thank you. We modified this sentence based on your comments to “Live attenuated anthrax vaccines were prepared using plasmid-cured strains.”(line 324)

Point 11: line 394: Extensive editing of references’ format is needed in order to comply with journal’s requirements.

Response 11: Thank you. We updated the citations and bibliography using the MDPI reference style in the Endnote software package.

Reviewer 2 Report

After reading the manuscript entitled "Genome Sequence and Phenotypic Analysis of a Protein Lysis-Negative, Attenuated Anthrax Vaccine Strain", I would like to make a few comments.

1) The main comment concerns the Inthroduction section.In my opinion, this section is written in insufficient detail. If the authors describe the history of the development and use of live vaccine strains, they should mention not only the Pasteur and Stern vaccines, but also other live vaccines. Also in this section, the authors did not give the reader a clear understanding of why they even carried out this study.

2) Lines 23 and  41

"Bacillus anthracis is an ancient Gram-positive bacterium" 

This is not a very good wording, since Bacillus anthracis is considered to be a relatively young species. Considering that the question of the age of this species is irrelevant to the subject of this manuscript, I propose to remove the wording "ancient"

3) Line 27 

PNO2 (PNO2D1) was Gram-positive

This is an obvious thing. It has already been stated in the text that the species Bacillus anthracis is Gram-positive, it is not necessary to repeat this for each individual strain.

4) Lines 56, 84, 87, 109, 147, 284, 334, 353, 359

Reference numbers are written in superscript format.

5) Line 95, subsection 2.2 Phenotypic characterization

The authors did not describe here how they determined lecithinase and hemolytic activity, however, they described the results of studying these phenotypic properties below.

6) Lines 164-165 

PNO2D1 sporulation ability appeared much weaker than that of A16Q1

This is too subjective and not precise enough. Could the authors provide some quantitative indicators here?

7) Lines 165-166 

Both A16Q1 and PNO2D1 strains were Gram-positive 

This phrase seems redundant, since all B.anthracis strains  are Gram-positive, there is no reason to expect this property to change.

8) Lines 198-201

Interestingly, genome-wide single nucleotide polymorphism (SNP) genotyping results showed that both Cvac02 and PNO2D1 were more closely related to Tsiankovskii strains than to Pasteur strains ATCC240 and ATCC4728 and Smith strains

I believe that this statement should be illustrated with a dendrogram, and there should be a link to it here.

9) Lines 214-215

pXO1 of the Pasteur II strain reported by Liang et al. (GenBank: gb|KT186230.1) indicated that pXO1 may also have originated from the Tsiankovskii strain

With all due respect to the authors, I would call this statement controversial and overly speculative. It suggests the possibility of exchanging pXO plasmids between different strains. As far as I know, this phenomenon has not yet been described. Since the statement that is the subject of this comment is not important to the manuscript, I believe that it should be removed from the text.

10) Lines 214-215

Inspection of the PNO2 and PNO2D1 genomes revealed a 7-base insertion mutation (CGAAAAT) within the nprR gene 

I suppose it would be better to indicate the mutation coordinates here

Author Response

Responses to Reviewer 2

Thank you for your letter and comments concerning our manuscript. The comments were valuable and helpful for revising and improving our paper. We have reviewed the comments carefully and made the necessary corrections. The revised portions are in red font in the revised manuscript. The responses to the comments are as follows.

Point 1: The main comment concerns the Inthroduction section.In my opinion, this section is written in insufficient detail. If the authors describe the history of the development and use of live vaccine strains, they should mention not only the Pasteur and Stern vaccines, but also other live vaccines. Also in this section, the authors did not give the reader a clear understanding of why they even carried out this study.

Response 1: A1: Thank you for your comments. We added the relevant content to the Introduction (lines 62-74).

Point 2: Lines 23 and 41

"Bacillus anthracis is an ancient Gram-positive bacterium" 

This is not a very good wording, since Bacillus anthracis is considered to be a relatively young species. Considering that the question of the age of this species is irrelevant to the subject of this manuscript, I propose to remove the wording "ancient"

Response 2: Thank you for your suggestion. Bacillus anthracis was shown to be the etiological agent of anthrax at the end of the nineteenth century. This marked the beginning of medical microbiological research. It was in this sense that we used the word “ancient”. Because this created ambiguity, we removed the word “ancient” as per your comment. (line 35)

Point 3: Line 27 

PNO2 (PNO2D1) was Gram-positive

This is an obvious thing. It has already been stated in the text that the species Bacillus anthracis is Gram-positive, it is not necessary to repeat this for each individual strain.

Response 3: Thank you for carefully reading our manuscript. Because PNO2 exhibits some atypical phenotypes for B. anthracis, we made some basic staining observations. Because the description of it as "Gram-positive" in the abstract seemed redundant, we removed it.

Point 4: Lines 56, 84, 87, 109, 147, 284, 334, 353, 359

Reference numbers are written in superscript format.

Response 4: Thank you. We updated the citations and bibliography using the MDPI reference style in the Endnote software package.

Point 5: Line 95, subsection 2.2 Phenotypic characterization

The authors did not describe here how they determined lecithinase and hemolytic activity, however, they described the results of studying these phenotypic properties below.

Response 5: Thank you for your comments. We added the relevant content to subsection 2.2 (lines 108–113).

Point 6: Lines 164-165 

PNO2D1 sporulation ability appeared much weaker than that of A16Q1

This is too subjective and not precise enough. Could the authors provide some quantitative indicators here?

Response 6: This is indeed a subjective description. Thus, we added At 24 h, A16Q1 was nearly fully sporulated, whereas PNO2D1 was still mostly in the vegetative state (Figure 1A)” (lines 175–177). We also examined the sporulation rates (Figure 3D). We apologize for not performing comparative and statistical analyses of the sporulation rates between A16Q1 and PNO2D1. We improved this in the revised manuscript (lines 247–248 and Figure 3D).

Point 7: Lines 165-166 

Both A16Q1 and PNO2D1 strains were Gram-positive 

This phrase seems redundant, since all B.anthracis strains  are Gram-positive, there is no reason to expect this property to change.

Response 7: We agree with you and have removed “Gram-positive”.

Point 8: Lines 198-201

Interestingly, genome-wide single nucleotide polymorphism (SNP) genotyping results showed that both Cvac02 and PNO2D1 were more closely related to Tsiankovskii strains than to Pasteur strains ATCC240 and ATCC4728 and Smith strains

I believe that this statement should be illustrated with a dendrogram, and there should be a link to it here.

Response 8: Thank you for your comment. Figure 2A is now linked here (line 210).

Point 9: Lines 214-215

pXO1 of the Pasteur II strain reported by Liang et al. (GenBank: gb|KT186230.1) indicated that pXO1 may also have originated from the Tsiankovskii strain

With all due respect to the authors, I would call this statement controversial and overly speculative. It suggests the possibility of exchanging pXO plasmids between different strains. As far as I know, this phenomenon has not yet been described. Since the statement that is the subject of this comment is not important to the manuscript, I believe that it should be removed from the text.

Response 9: Thank you. We agree with your comments. The reported Pasteur II strain (containing two plasmids) does not have a published full genome sequence, and the Pasteur-type strains from the NCBI database lack the pXO1 plasmid. Thus, the clustering analysis was not comprehensive, and the extrapolation was too arbitrary. Therefore, we rewrote this section (line 224). “Full-sequence SNP clustering analysis results for pXO1 of the Pasteur II strain reported by Liang et al. (GenBank: gb|KT186230.1) showed that it was also similar to pXO1 of the Tsiankovskii strain (Figure 2B).”

Point 10: Lines 214-215

Inspection of the PNO2 and PNO2D1 genomes revealed a 7-base insertion mutation (CGAAAAT) within the nprR gene 

I suppose it would be better to indicate the mutation coordinates here

Response 10: Thank you. We rewrote this as “The PNO2 and PNO2D1 genomes each contained a 7-base insertion mutation (CGAAAAT) within the nprR gene position between bases 123 and 124.” (line237-238)

Round 2

Reviewer 1 Report

The authors responded comprehensively to most of the suggestion and questions. The manuscript can be accepted for publication.   Here are a few final less-important notes. Indeed Bacillus anthracis is an ancient bacterium. The disease impact has been described in ancient literature dating back more than 2000 years. It is even described in the book of "GENESIS". Even its name is ancient. Bacillus anthracis, the organism that causes anthrax, derives its name from the ancient Greek word for coal (anthrax) because of its ability to cause black, coal-like cutaneous eschars.  About  the possibility of exchanging pXO plasmids between different strains. This is also true, bibliography describes this.

Author Response

Thank you very much for your comments! We also agree with you, which is why we used the word "ancient". As reviewer 2 commented, B. anthracis is considered to be relatively young in terms of origin. Also, from the origin point of view, the origin of B. anthracis must have been accompanied by the transfer of pXO plasmids in the B. cereus group. However, in order not to cause ambiguity, we have made this revision.

In addition, we noted that the method section could be improved. We have modified Line166-175.

Reviewer 2 Report

In my opinion, the authors of the manuscript entitled "Genome Sequence and Phenotypic Analysis of a Protein Lysis-Negative, Attenuated Anthrax Vaccine Strain" have made considerable efforts to edit their manuscript in accordance with the comments of the reviewers. I have no comments on the current version of this manuscript, and I believe that it could be published.

Author Response

Thank you. We appreciate the reviewers very much for their constructive comments and suggestions on our manuscript.